# Camera Pose Estimation Emerging in Video Diffusion Transformer

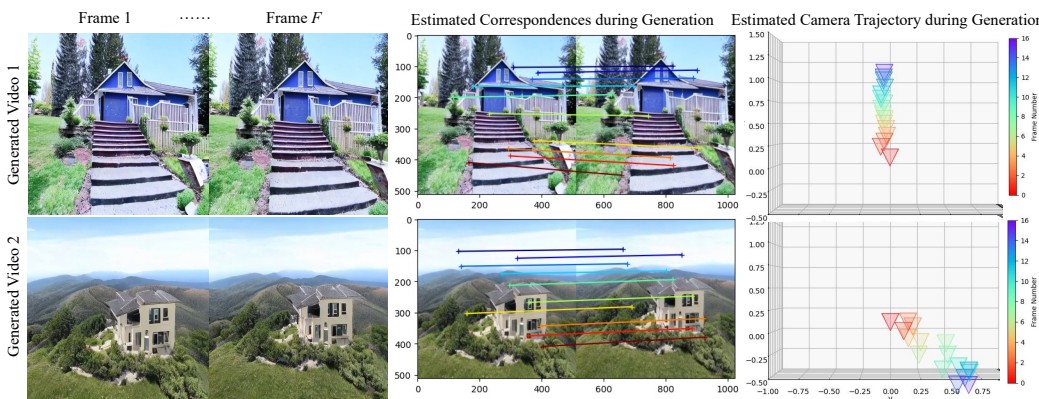

Figure 1: **JOG3R** creates realistic videos of stationary scenes while *simultaneously* generating the associated camera pose for each frame. Please refer to the supplementary page for video results.

## Abstract

Diffusion-based video generators are now a reality. Being trained on a large corpus of real videos, such models can generate diverse yet realistic videos (Brooks et al., 2024; Zheng et al., 2024). Given that the videos appear visually coherent across camera changes, we ask, *do the underlying generators implicitly learn camera registrations?* Hence, we propose a novel adaptation to repurpose the intermediate features of the generator for camera pose estimation by linking them to the SoTA camera calibration decoder of DUSt3R (Wang et al., 2024a). This effectively unifies the video generation and camera estimation into a single framework. On top of unifying two different networks into one, our architecture can directly be trained on real video and simultaneously produces correspondence, with respect to the first frame, for all the video frames. Our final model, named JOG3R can be used in text-to-video mode, and additionally it produces camera pose estimates at a quality on par with the SoTA model DUSt3R, which was trained exclusively for camera pose estimation. We report that the synergy between video generation and 3D camera reconstruction tasks leads to around 25% better FVD scores with JOG3R against pretrained OpenSora.

## 1 Introduction

Video diffusion models have rapidly improved over the last two years, leading to the emergence of many commercial and open-sourced models (Guo et al., 2024; Zheng et al., 2024; Brooks et al., 2024; Menapace et al., 2024a; Blattmann et al., 2023a). They are trained on very large-scale datasets, *e.g.*, WebVid10M (Bain et al., 2021) or Panda-70M (Chen et al., 2024), and produce realistic, diverse, and temporally smooth videos, simply based on text or image prompts.

In another recent breakthrough, DUSt3R (Wang et al., 2024a) demonstrated that the long-standing optimization-based structure-from-motion framework for camera estimation can be directly replaced by the forward pass of a dedicated network that has been trained to establish correspondence between any given pair of video frames. This is in contrast to the current SoTA in optimization-based approach for structure-from-motion GLOMAP (Pan et al., 2024a).

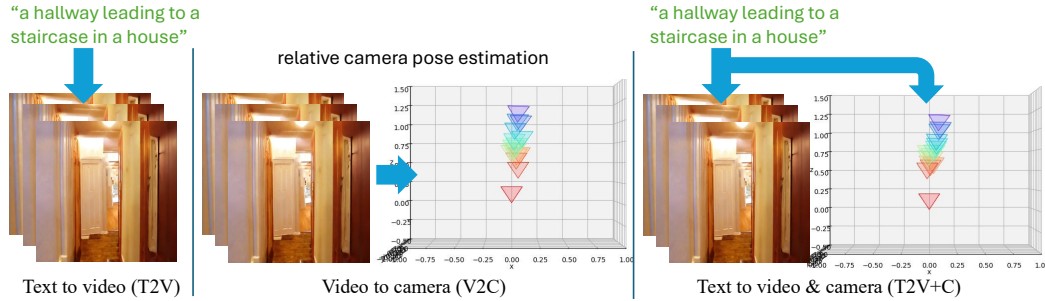

Text to video (T2V)   Video to camera (V2C)   Text to video & camera (T2V+C)

Figure 2: JOG3R is a versatile model that can (a) generate a video from text, (b) reconstruct 3D camera motion given a video, and (c) generate a video and the corresponding camera motion simultaneously. The camera trajectories obtained in (b) and (c) are consistent.

Inspired by the emergent behavior of intermediate features of large-scale image generators towards other tasks (*e.g.*, correspondence, semantic segmentation, etc. Tang et al. (2023); Dutt et al. (2024)), we ask if the pretrained video generator features have similar emergent behavior. In particular, we investigate whether the pretrained features can be repurposed towards DUSt3R-like camera pose estimation. Surprisingly, we find that the video generator features, OpenSora in our setting, do not natively have emergency behavior and cannot be used directly for camera tracking.

Instead, we investigate whether the video generator features can be adapted towards camera pose estimation. In particular, we test if with a limited amount of fine-tuning, one can produce video generator features that also can be reused for camera tracking, without sacrificing video generation quality (see Figure 1). We present a *JOint Generation and 3d camera Reconstruction* network, in short JOG3R, that combines video generation with camera pose estimation into a single network, and can be supervised with generation and 3D reconstruction losses. We demonstrate that this does not lead to a loss in video quality while setting a new SoTA with respect to camera tracking on real video using a feedforward network (see Figure 2). In fact, we find that training with camera reconstruction leads to improved video generation, leading to a notable improved FVD score on the RealEstate10K-test.

In summary, the paper makes the following contributions:

- The first model that can both generate videos and estimate 3D cameras;

- Extensive experiment and study on how well the video features can be used for 3D camera estimation and ablating the various design choices; and

- Reporting SoTA video-based camera tracking results on both RealEstate10k-test and DL3DV10K datasets.

## 2 RELATED WORK

### 2.1 DIFFUSION-BASED VIDEO GENERATION

Building on the success of diffusion models (Ho et al., 2020; Song et al., 2020) in image synthesis (Dhariwal & Nichol, 2021; Rombach et al., 2021), the research community has extended diffusion-based methods to video generation. Early works (Ho et al., 2022a;b) adapted image diffusion architectures by incorporating a temporal dimension, enabling the model to be trained on both image and video data. Typically, U-Net-based architectures incorporate temporal attention blocks after spatial attention blocks and 2D convolution layers are expanded to 3D convolution layers by altering kernels (Ho et al., 2022b; Wu et al., 2023). Latent video diffusion models (Blattmann et al., 2023b; He et al., 2022; Wang et al., 2023b; Blattmann et al., 2023a) have been introduced to avoid excessive computing demands, implementing the diffusion process in a lower-dimensional latent space. Seeking to generate spatially and temporally high-resolution videos, another line of research adopts cascaded pipelines (Ho et al., 2022a; Singer et al., 2022; Zhang et al., 2023a; Wang et al., 2023c;

Bar-Tal et al., 2024), incorporating low-resolution keyframe generation, frame interpolation, and super-resolution modules. To maximize computational scalability, recent waves in video generation (Chen et al., 2023; Ma et al., 2024; Menapace et al., 2024b; Brooks et al., 2024; Zheng et al., 2024) diverge from U-Net-based architecture and employ Diffusion Transformer (DiT) (Peebles & Xie, 2023) backbone that processes space-time patches of video and image latent codes. Following this direction, we build our method on OpenSora (Zheng et al., 2024), a publicly available DiT-based latent video diffusion model.

## 2.2 3D Reconstruction

The fundamental principles of multiview geometry Wrobel (2001) including feature extraction Lowe (2004); Brown et al. (2011), matching Agarwal et al. (2009); Lou et al. (2012); Wu (2013a); Havlena & Schindler (2014), and triangulation with epipolar constraints are well known to produce highly accurate (yet spare) 3D point clouds with precise camera pose estimation from multiview images Schonberger & Frahm (2016). The efficiency of 3D reconstruction has been improved with linear-time incremental structure-from-motion Wu (2013b) and coarse-to-fine hybrid approaches Crandall et al. (2012); Cui et al. (2017). To improve robustness to outliers, researchers proposed global camera rotation averaging Cui et al. (2017), camera optimization techniques based on features of points vanishing with oriented planes Holynski et al. (2020) or from a learned neural network Lindenberger et al. (2021) to prevent rotation and scale drift issues in the process of the structure-from-motion. Global camera pose registration and approximation with geometric linearity Jiang et al. (2013); Cai et al. (2021) or joint 3D point position estimation Pan et al. (2024a) are designed to further push the scalability and efficiency of the 3D reconstruction as well as the robustness particularly to the image sequence with small baselines.

Given estimated camera poses and sparse 3D point clouds, multiview stereo can then produce a dense 3D surface using hand-created visual features Schönberger et al. (2016) or neural features with a cost volume Ma et al. (2022); Ummenhofer & Koltun (2021); Ma et al. (2022); Zhang et al. (2023c); Ye et al. (2023) to predict globally coherent depth estimates. Existing neural rendering methods reconstruct such a dense surface by modeling the implicit or explicit cost volume and differentiable rendering of the scene for photometric supervision from multiview images Li et al. (2023b); Sun et al. (2022); Peng et al. (2023); Guo et al. (2022); Yu et al. (2022); Wang et al. (2022); Oechsle et al. (2021); Wang et al. (2021); Murez et al. (2020) or monocular depth estimation Sayed et al. (2022). Some pose-free methods further erase the requirement of camera calibration: test time optimization produces globally consistent depth map under unknown scale and poses using frozen depth prediction model Xu et al. (2023); the unsupervised signals from dense correspondences such as optical flow is integrated to learn from unlabeled data Yin & Shi (2018); Teed & Deng (2018); Zhou et al. (2019). Recent works proposed a direct regression framework for dense surface reconstruction from pairwise images by learning to predict globally coherent depths and camera parameters Ummenhofer et al. (2016) or to directly predict per-pixel 3D point clouds from two views Wang et al. (2024b); Leroy et al. (2024) using a vision transformer with dense tokenization techniques Ranftl et al. (2021).

## 2.3 Diffusion Model as Features for 3D Reconstruction

A generative diffusion model is often trained on millions of paired image and text prompts and in the process develops a semantically meaningful visual prior. Naturally, researchers are interested in using this strong prior for many downstream 3D vision tasks. Injecting 3D awareness into the diffusion prior greatly improves the accuracy and generalizability of the monocular depth estimation and correspondence search tasks El Banani et al. (2024); Yue et al. (2024). The latent features from the frozen pretrained diffusion model are often used as a backbone, and a task-specific decoder with cross attention is newly trained for semantic correspondences Tang et al. (2023); Zhang et al. (2023b); Hedlin et al. (2024); Zhang et al. (2024); Hedlin et al. (2024); Jiang et al. (2024), 3D correspondences Dutt et al. (2024), semantic segmentation and monocular depth estimation Zhao et al. (2023), material and shadow prediction Zhan et al. (2023), general object 3D pose estimation Örnek et al. (2023); Cai et al. (2024). However, such image diffusion features do not inherently consider the temporal relation between the frames, leading to temporally unstable 3D prediction results from videos. In contrast, we propose to utilize the video diffusion features as a backbone for the multitasking prediction of video generation and 3D camera poses estimation.

# 3 METHOD

## 3.1 MODEL AND PRELIMINARIES.

**Video diffusion model.** We consider OpenSora (Zheng et al., 2024) as our base video generation model, which is a DiT-based video generator inspired by the impressive success of Sora (Brooks et al., 2024). OpenSora performs the diffusion process in a lower-dimensional latent space defined by a pre-trained VAE encoder $\mathcal{E}$. Each frame $x$ of the input video is first projected into this latent space, $z_0 = \mathcal{E}(x)$. Given a diffusion time step $t$, the *forward* process incrementally adds Gaussian noise to the latent code $z_0$ via a Markov chain and obtains noisy latent $z_t$. The denoising model $\epsilon_\theta$ takes the noisy latents of all frames, the time step $t$, and the text prompt $y$ as input to predict the added noise: $\epsilon_\theta(\{z_t^f\}_{f=1}^F, t, y)$, where $F$ is the total number of frames and $\theta$ denotes the parameter of the DiT network (Peebles & Xie, 2023). The network consists of $m+1$ spatial-temporal diffusion transformer (STDiT) blocks $\{b^0, \ldots, b^m\}$, similar to Ma et al. (2024). The iterative process of noise prediction and noise removal is referred to as the *backward* process.

**Camera pose estimation module.** We employ the state-of-the-art multi-view stereo reconstruction (MVS) framework DUSt3R (Wang et al., 2024a) as our downstream camera tracking branch. Given a pair of images, DUSt3R first encodes each one individually with a ViT encoder (Dosovitskiy et al., 2021; Weinzaepfel et al., 2022). A pair of decoders take both features as input for cross-view information sharing, followed by two separate heads estimating point maps $X \in \mathbb{R}^{H \times W \times 3}$ represented in the coordinate of the first view, denoted as $X^{1,1}$ and $X^{2,1}$, respectively. The relative camera pose is then estimated by aligning $X^{1,1}$ and $X^{1,2}$ (or, equivalently $X^{2,1}$ and $X^{2,2}$) using Procrustes alignment (Luo & Hancock, 1999) with PnP-RANSAC (Lepetit et al., 2009; Fischler & Bolles, 1981).

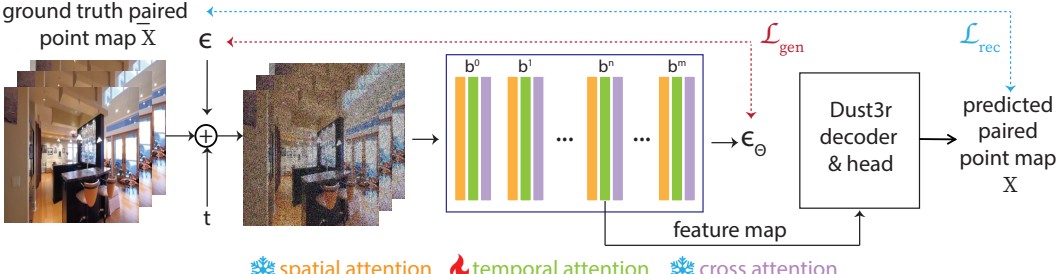

Figure 3: JOG3R repurposes the intermediate features from a video generation model for camera pose estimation by routing them to the SoTA camera calibration decoder of DUSt3R. We train both the temporal layers of the generation model as well as the DUSt3R decoders using a combination of generation and reconstruction losses.

## 3.2 JOINT GENERATION AND RECONSTRUCTION DiT NETWORK

We propose a unified network that is able to do both video denoising and camera tracking. We observe that ViT and DiT actually share many architectural designs in common since they both belong to the broad transformer family. Hence, our key insight is to replace the *image*-based ViT encoder in DUSt3R with the *video* DiT backbone in OpenSora. In other words, we provide the features of the denoising DiT network $\epsilon_\theta$ to DUSt3R decoders and heads, see Figure 3 for illustration.

Specifically, we extract the output of the intermediate STDiT block $b^n$ at a particular time step $t$ during the backward process. Following Tang et al. (2023), we consider small $t$ where the feature focuses more on low-level details, making it useful as a geometric feature descriptor to build correspondence across frames.

**Our modification of DUSt3R.** The features extracted from the video generator encode a sequence of $F$ frames and are provided to the DUSt3R decoders. During training, we sample a pair of frames $\{(1, f)\}_{f=2}^F$ to predict the 3D point maps between the first frame and any other frame $f$ in the

sequence. At inference time, we first predict the point maps between all pairs $(1, f)$ and perform a global camera registration to obtain the camera pose estimation for the whole sequence.

**Training objectives.** During training, our model is supervised by two objectives: generation loss $\mathcal{L}_{\text{gen}}$ and reconstruction loss $\mathcal{L}_{\text{rec}}$. The generation loss $\mathcal{L}_{\text{gen}}$ is the common objective in training diffusion models that aims to match the added noise $\epsilon$. The reconstruction loss $\mathcal{L}_{\text{rec}}$, following the definition in DUSt3R, is the sum of confidence-weighted Euclidean distance $L_2(f, i)$ between the regressed point maps $X$ and the ground truth point maps $\bar{X}$ over all valid pixels $i$ and all frames $f$. Formally,

$$\mathcal{L}_{\text{gen}} = \left\| \epsilon - \epsilon_\theta \left( \{z_t^f\}_{f=1}^F, t, y \right) \right\|_2^2 \tag{1}$$

$$\mathcal{L}_{\text{rec}} = \sum_{f \in \{2,..,F\}} \sum_i C_i^{f,1} L_2(f, i) - \alpha \log C_i^{f,1} \tag{2}$$

$$L_2(f, i) = \left\| \frac{1}{s} X_i^{f,1} - \frac{1}{\bar{s}} \bar{X}_i^{f,1} \right\|_2$$

where the scaling factors $s$ and $\bar{s}$ handles the scale ambiguity between prediction and ground-truth by bringing them to a normalized scale, $C_i^{f,1}$ is the confidence score for pixel $i$ which encourages network to extrapolate in harder areas, and $\alpha$ is a hyper-parameter controlling the regularization term (Wan et al., 2018). We refer interested readers to Wang et al. (2024a) for more details. The final loss is defined as $\mathcal{L}_{\text{total}} = \mathcal{L}_{\text{gen}} + \lambda \mathcal{L}_{\text{rec}}$, and we empirically set $\lambda = 1$.

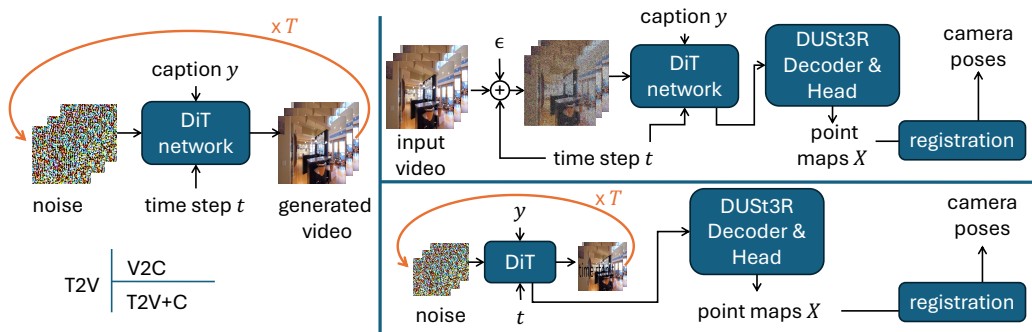

Figure 4: JOG3R supports text-to-video (T2V), video to camera estimation (V2C), and joint video generation and camera estimation (T2V+C) at inference time.

**Inference.** Once trained, JOG3R naturally supports three ways of inference (see Figure 2 and supplemental video): (i) *Text-to-video (T2V)*: the input is sampled Gaussian noise and we iteratively denoise it with the text guidance to generate a video. (ii) *Video-to-camera (V2C)*: we add noise to the input video based a sampled time step $t$, denoise it for one time step, route the feature maps to DUSt3R decoders and heads, followed by registration of point maps $X$ to obtain camera poses.

Given the two inference modes above, a straightforward combination is using the generated video of T2V as the input of V2C, which we denote as T2V→V2C, essentially chaining the two networks. However, thanks to our novel network design, we can provide the feature map directly to the reconstruction module at the desired time step, without the overhead of adding noise and passing it through the network again. As a result, cameras are generated alongside the video *in one go*. We term such a tightly coupled joint inference mode as (iii) *Text-to-Video+Camera (T2V+C)*. Fig. 4 illustrates the pipeline of these three inference modes.

**Implementation Details.** We adopt OpenSora 1.0 as our video generator, which uses 2D VAE (from Stability-AI) Rombach et al. (2022), T5 text encoder (Raffel et al., 2020), and an STDiT (ST stands for spatial-temporal) architecture similar to variant 3 in Ma et al. (2024) as the denoising network. Among the 28 STDiT blocks, we empirically set the first 4 frozen and update only the weights of the temporal attention layers for the remaining 24 blocks. We extract the output of the 26[th] block $b^{25}$

as feature maps for DUSt3R decoders. The final two blocks behave as a "generation" branch whose weights are only updated by the gradient of generation loss $\mathcal{L}_{\text{gen}}$.

We adopt the linear prediction head of DUSt3R for final pointmap estimation. DUSt3R originally uses a decoder with 12 transformer blocks that is duplicated for each of the pair of frames. However, information sharing is enabled between the two decoders. In our experiments, we find that a decoder structure with six transformer blocks provides similar performance and report our results accordingly. Furthermore, since the features we get from the generator encode all the frames in a video sequence, we also experiment with replacing the duplicate decoder architecture with a single decoder consisting of 6 transformer blocks that perform full 3D attention across all the frames. We empirically find that this performs on par with duplicate decoders (see Table 1), and hence we use the latter to provide a more fair comparison to DUSt3R.

During training, we sample the time step $t \in [0, 10]$ (corresponding to 10% of noise level) and consider the empty prompt for computing the reconstruction loss $\mathcal{L}_{\text{rec}}$, while for the generation loss $\mathcal{L}_{\text{gen}}$ we sample the full range of time steps and use the captions of the videos. At test time, we sample $t \in [0, 5]$ to add noise to the input video for camera estimation (V2C). To perform joint camera estimation and video generation (T2V+C), we run the standard T2V pipeline of OpenSora and when the time step hits the sampled $t \in [0, 5]$, we provide the output of block $b^{25}$ to DUSt3R for camera estimation.

## 4 EXPERIMENTS

In this section, we evaluate the proposed method in three aspects. We follow standard approaches to assess the generated video quality (T2V). Since there is no ground truth camera trajectories for the videos generated from T2V+C, we focus on validating the accuracy of camera pose estimation on real videos (V2C) and report self-consistency for T2V+C.

### 4.1 SETUP

**Data.** We choose RealEstate10K Zhou et al. (2018) as the dataset, which has around 65K video clips paired with camera parameter annotations. We use the captions of RealEstate10K provided in He et al. (2024) and also follow their train/test split. As pre-processing, we pre-compute the VAE latents of the video frames and the T5 text embeddings of the captions. To obtain point map annotation $\bar{X}$, we first estimate metric depth with ZoeDepth (Bhat et al., 2023), un-project it to 3D and transform to the coordinate of the first frame using the camera parameters provided in RealEstate10K. All camera extrinsic parameters are expressed with respect to the first frame.

In addition, we consider DL3DV10K (Ling et al., 2024), which also provides camera annotations, as a failed test set. We choose a random set of 70 videos for testing and caption the first frame of each video using Li et al. (2023a). We prepare point map annotations using ZoeDepth (Bhat et al., 2023), similar to the RealEstate10K dataset.

**Baselines.** We compare with a pair-wise method DUSt3R Wang et al., 2024a with linear head and a video-based SfM method GLOMAP (Pan et al., 2024b). For DUSt3R we consider two variants: (i) off-the-shelf pretrained weights (DUSt3R$^{\dagger}$) and (ii) trained from scratch with the same data as ours (DUSt3R*). For GLOMAP we report the results before the global bundle adjustment part.

**Metric.** We validate the quality of camera tracking on real videos (V2C) by comparing the estimated camera poses $(\mathbf{R}, \mathbf{t})$ with the ground truth poses $(\bar{\mathbf{R}}, \bar{\mathbf{t}})$. For rotation, we compute the relative error angle between two rotation matrices. Since the estimated and ground truth translation can differ in scale, we follow Wang et al. (2023a) to compute the angle between the two normalized translation vectors, *i.e.*, $\arccos(\mathbf{t}^{\top}\bar{\mathbf{t}}/(\|\mathbf{t}\|\|\bar{\mathbf{t}}\|))$. Besides reporting the average of the two errors, we also follow Wang et al. (2024a) to report Relative Rotation Accuracy (RRA) and Relative Translation Accuracy (RTA), *i.e.*, the percentage of camera pairs with rotation/translation error below a threshold. Due to limit of the number of frames, hence small rotation variations, we select a threshold $5°$ to report RTA@5 and RRA@5. Additionally, we calculate the mean Average Accuracy (mAA@30), defined as the area under the curve accuracy of the angular differences at min(RRA@30, RTA@30). We also use FID (Heusel et al., 2017) and FVD (Unterthiner et al., 2019) to measure image and video quality respectively, ensuring that our method maintains high generation quality.

| Method | Rot. err. (°) ↓ | Transl. err. (°) ↓ | RRA@5 ↑ | RTA@5 ↑ | mAA@30 ↑ |
|---|---|---|---|---|---|
| (0) ours w/ 3D attn | 0.38 | 36.86 | 99.49% | 9.17% | 32.15% |
| (1a) ours w/o $\mathcal{L}_{\text{gen}}$ | 0.36 | 33.09 | 99.71% | 12.18% | 34.55% |
| (1b) JOG3R (ours) | 0.37 | 32.66 | 99.77% | 13.16% | 35.62% |
| (2a) DUSt3R$^\dagger$ | 0.77 | 36.61 | 97.56% | 7.54% | 30.13% |
| (2b) DUSt3R* | 0.33 | 30.51 | 99.71% | 12.76% | 37.88% |
| (3) GLOMAP | 0.96 | 19.55 | 96.86% | 25.92% | 55.82% |

Table 1: **V2C error comparison on RealEstate10K-test.** DUSt3R$^\dagger$ indicates pretrained DUSt3R weights, whereas DUSt3R* is trained with the same training set as our method – RealEstate10K-train.

| Method | Rot. err. (°) ↓ | Transl. err. (°) ↓ | RRA@5 ↑ | RTA@5 ↑ | mAA@30 ↑ |
|---|---|---|---|---|---|
| (1a) ours w/o $\mathcal{L}_{\text{gen}}$ | 8.77 | 59.04 | 48.31% | 0.37% | 3.54% |
| (1b) JOG3R (ours) | 9.01 | 58.82 | 47.73% | 0.24% | 3.86% |
| (2a) DUSt3R$^\dagger$ | 10.27 | 61.91 | 46.82% | 0.33% | 2.91% |
| (2b) DUSt3R* | 8.38 | 58.70 | 49.78% | 0.33% | 3.93% |
| (3) GLOMAP | 10.57 | 62.97 | 46.62% | 0.21% | 2.60% |

Table 2: **V2C error comparison on DL3DV10K.**

## 4.2 Reconstruction Evaluation

In Table 1, we compare the camera pose estimation (V2C) errors on RealEstate10K-test and report the errors of withheld DL3DV10K in Table 2. Comparing (1a) and (1b) of two tables, we see that removing generation loss $\mathcal{L}_{\text{gen}}$ leads to overall worse results than our full model, confirming the hypothesis that *retaining generation ability helps reconstruction*.

Our full method – JOG3R, performs overall better than pretrained DUSt3R on both datasets, cf., (1b) and (2a). When trained with the same RealEstate10K-train, JOG3R still has on-par reconstruction quality compared with the DUSt3R counterpart DUSt3R*. When we replace the original DUSt3R decoders with full 3D attention blocks (0), we obtain on-par results with a marginal drop in accuracy.

We also report the results of GLOMAP (Pan et al., 2024b) before the final bundle adjustment step. It is the state-of-the-art method in a well studied SfM problem, which can be treated as a role of the upper bound to indicate how far we are. In Table 1 row (3), we observe it does surpass other methods in RealEstate10K, where videos often contain smaller motion and hence smaller baselines for each stereo pair. When the overlap between consecutive frames gets smaller, like in DL3DV10K, such a video-based method struggles and our method actually yields lower errors than GLOMAP.

Figure 5 shows the qualitative comparison of our method and baselines. Since camera poses are estimated through registration, which builds 3D correspondences along the way, we visualize the final camera trajectories as well as the correspondence between the first and the last frame. One can see that our method produces good camera trajectories similar to DUSt3R, which is a method tailored for reconstruction only, but no generation. In the last row we show a failure case where *both* our method and DUSt3R fail to estimate reasonable camera poses. We hypothesize this is due to the infinite depth in the sky region which could cause inconsistent scale normalization across each stereo pair.

## 4.3 Generation Evaluation

For each method, we generate 180 videos using the captions in RealEstate10K-test and report the FID/FVD against the real images/videos in RealEstate10K-test. Table 3 suggests that our full model generates more realistic images/videos than pretrained OpenSora ((1c) vs. (2)). When ablating the generation loss $\mathcal{L}_{\text{gen}}$, the quality slightly degrades compared to our full model ((1c) vs. (1a)). This is intuitive because without the generation loss, there is nothing to enforce the model to retain its

| Method | FID ↓ | FVD ↓ |
|---|---|---|
| (1a) ours w/o $\mathcal{L}_{\text{gen}}$ | 110.40 | 1898.72 |
| (1b) ours w/o $\mathcal{L}_{\text{rec}}$ | 88.02 | 1440.92 |
| (1c) JOG3R (ours) | 94.75 | 1339.74 |
| (2) pretrained OpenSora | 115.36 | 1872.41 |

Table 3: **Generation quality comparison.** We compute the FID and FVD with RealEstate10K-test.

full generation capability. See also supplemental videos. It is worth noting that (1b) corresponds to a baseline where $\mathcal{L}_{\text{rec}}$ is disabled by removing DUSt3R decoders/heads, *i.e.*, it is equivalent to standard video diffusion model finetuning except only the weights of the temporal attention layers are updated. We see removing $\mathcal{L}_{\text{rec}}$ leads to different impacts on FID and FVD. Since our method aims to generate videos, we argue FVD is a more important metric to measure the quality. As a result, the lower FVD of our full methdod (1c) suggests *learning camera pose estimation positively impacts the quality of video generation*. Figure 6 shows that our method generate realistic videos and the qualitative comparison also confirms the benefit of reconstruction loss $\mathcal{L}_{\text{rec}}$.

### 4.4 Discussion

**Synergy of two tasks.** Our full model JOG3R is trained with two losses, generation loss $\mathcal{L}_{\text{gen}}$ and reconstruction loss $\mathcal{L}_{\text{rec}}$. In both Table 1 and 2, (1a) and (1b), we show that keeping the generation loss $\mathcal{L}_{\text{gen}}$ helps the reconstruction branch attain better camera poses estimation. On the other hand, Table 3 (1b) and (1c) also suggest that introducing the reconstruction task results in better video generation quality. Empirically, we demonstrate a synergy between two tasks – learning to generate helps reconstruction; learning to reconstruct also helps generation. It shares the same spirit with the known "analysis and synthesis" analogy, but our architectural design tightly couples them in one network and allow end-to-end training.

**Self consistency of T2V→V2C and T2V+C.** Since one can use JOG3R to *generate* camera trajectories in two ways: cascading T2V and V2C or the tightly coupled T2V+C pipeline, it is worth comparing how much the two results differ. We run the two pipelines with 100 prompts and report $0.45°$ average difference in rotation and $19.20°$ in translation, both of which are low errors compared with the corresponding numbers in Table 1 and 2, indicating that the camera poses from joint T2V+C pipeline is consistent with T2V→V2C. The qualitative results in Figure 7 also confirm this conclusion.

## 5 Conclusions and Future Work

We have presented the first framework to enable joint video generation and 3D camera reconstruction. Our method utilizes intermediate features of a pre-trained video generation model for predicting relative 3D point maps and hence enabling camera registration. Specifically, by providing the intermediate generation features to task specific decoders and prediction heads, we present a unified framework for text-to-video generation (T2V), joint generation and camera estimation (T2V+C), and camera estimation for real videos (V2C).

While being first of its kind, our method is not without limitations. First of all, since it is not trivial to obtain accurate camera annotations for dynamic scenes, our method is currently trained and applicable for videos of static scenes only. The length of the video sequences our method can handle is currently limited by the number of frames the generator can synthesize. Handling longer sequences may require extending our method to operate in a sliding window manner. As the video generators continue to improve to enable generation of longer sequences, our method will also naturally extend to handling longer videos with larger baseline.

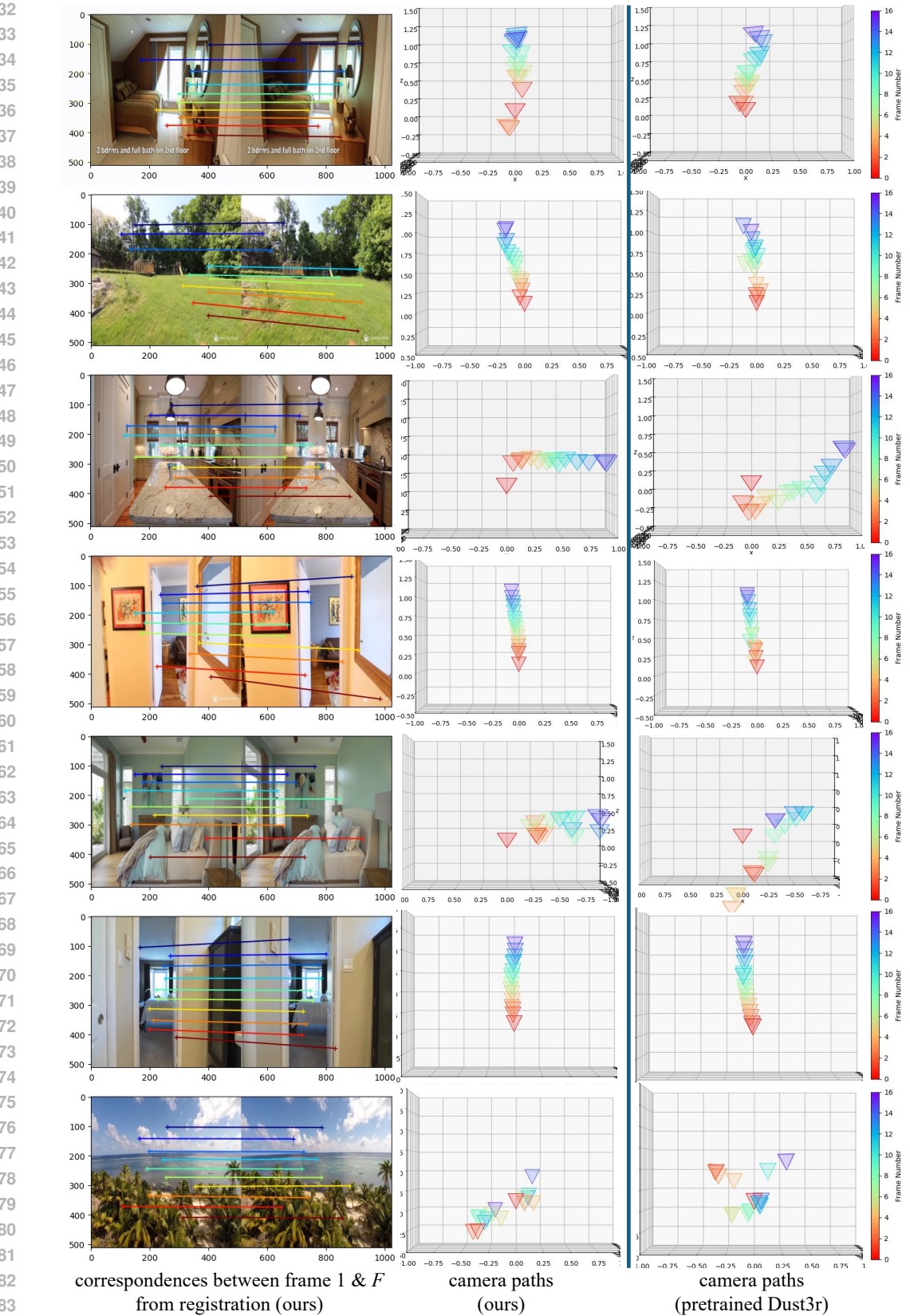

Figure 5: **Qualitative camera pose estimation (V2C) results.** The last row indicates a failure case. Please see suppmat. for videos and more analysis.

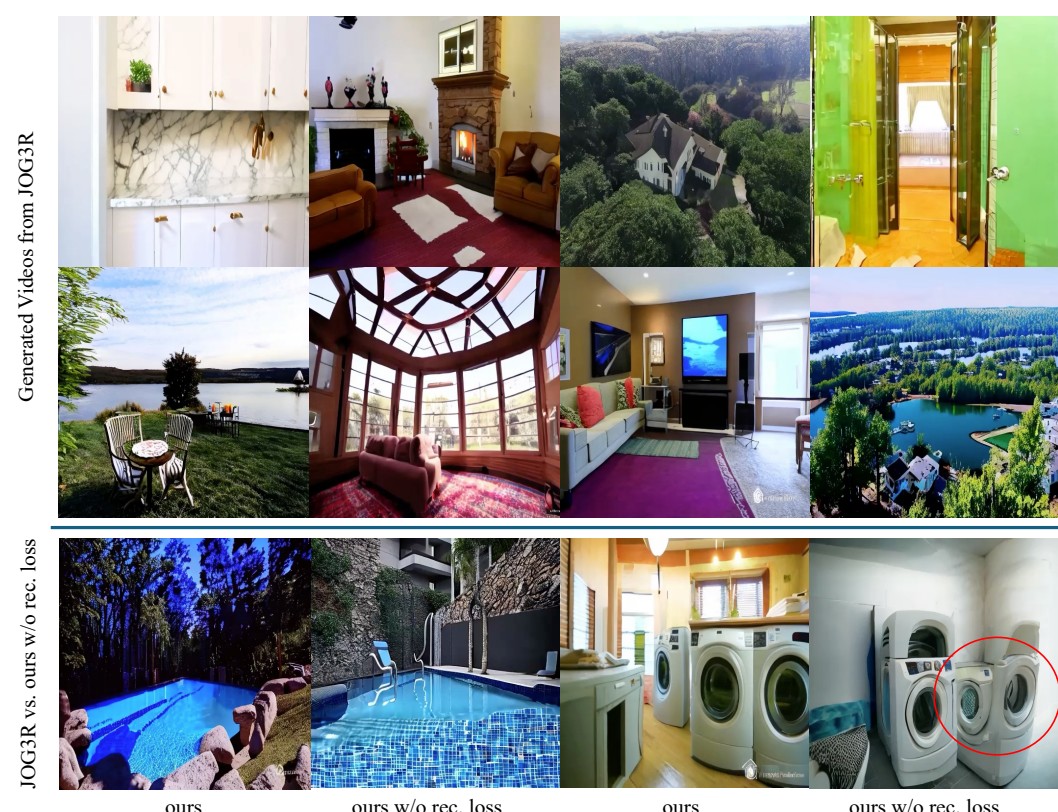

Figure 6: **Qualitative generation T2V results.** Please see suppmat. for videos.

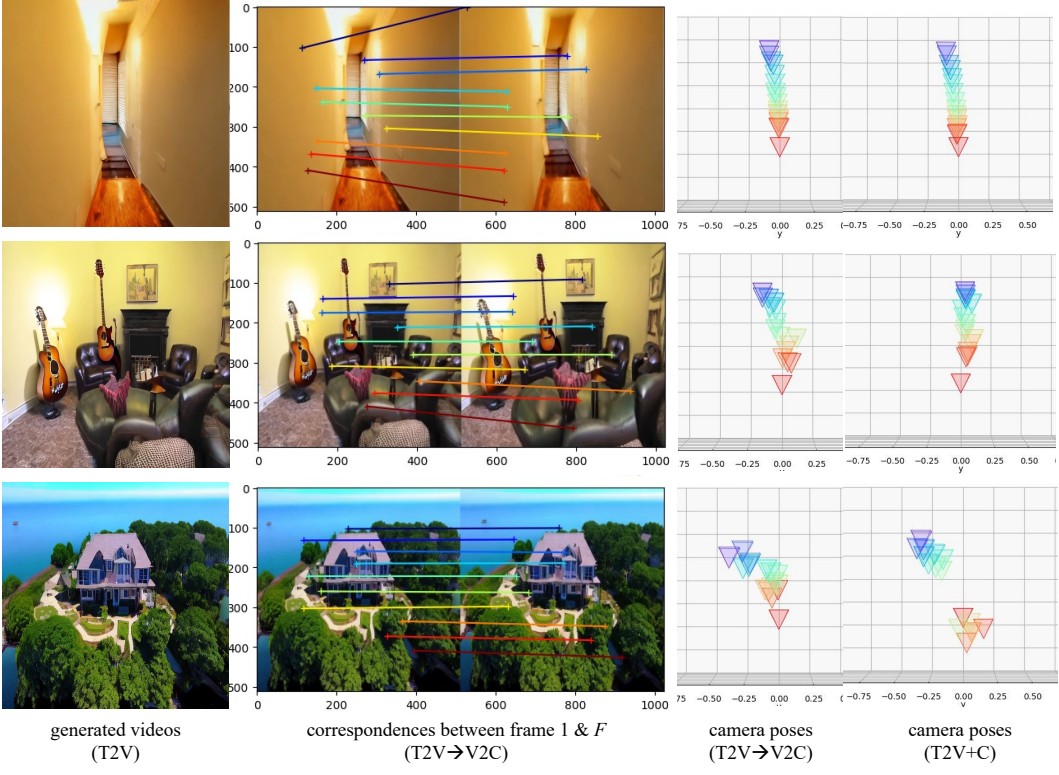

Figure 7: **Qualitative generation T2V+C results.** Please see suppmat. for videos.

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

# A APPENDIX

You may include other additional sections here.

