# OpenReview forum: "Camera Pose Estimation Emerging In Video Diffusion Transformer"
_ICLR.cc/2025/Conference — ICLR 2025 Conference Withdrawn Submission_

### Official Review · Reviewer_bNED · 2024-10-27

**Soundness:** 2
**Presentation:** 3
**Contribution:** 2
**Rating:** 3
**Confidence:** 4

**Summary:**

Summary:

This paper presents a method that generates video output alongside corresponding camera parameters. The method leverages video features within a video diffusion network, utilizing these features as input to the DUSt3R decoder to predict a dense 3D map, thereby estimating the camera pose.

**Strengths:**

Strengths:
good writing, easy to follow.
The concept is promising, especially since obtaining consistent camera poses during video diffusion could substantially enhance performance.

**Weaknesses:**

Weaknesses:
The integration of the OpenSORA framework with DUSt3R appears somewhat simplistic. The primary modification in the proposed method involves using video features at a single timestamp as an additional input for DUSt3R, which limits the optimization of the camera pose during inference. Furthermore, as noted, the timestamps ( t ) are sampled from ranges of 0-10 and 0-5 during training and testing, respectively. This small variation results in noisy images that closely resemble the clean image (at ( t=0 )). Consequently, the proposed method may not be able to compete with T2V+DUST3R*, as indicated by the results in Table 2.

Several experimental definitions are lacking, which constitutes a significant issue with the proposed method. The authors do not provide any details regarding the configuration of DUSt3R in the experimental section. For instance, DUSt3R can refine the depth map and camera pose via an Adam optimizer as part of a global refinement process, akin to bundle adjustment (BA), which can greatly improve camera accuracy. Based on the data presented in Table 1 and the descriptions in the main text, it appears that the authors did not use such a global refinement. Additionally, there is insufficient detail regarding the configuration of GLOMAP. It is crucial to note that GLOMAP is not a video-SfM method; rather, it is a global structure-from-motion technique. Thus, the claims made by the authors in the main text are misleading. Additionally, the authors' assertion that they "compared the results of GLOMAP before global BA" is unfair; removing global refinement would lead to a dramatic decrease in GLOMAP's performance. Even with the reported results prior to BA, the data in Tables 1 and 2 suggest that GLOMAP still achieves comparable results to the proposed method.

The absence of key baselines undermines the validity of the proposed method. Given that the authors propose generating both video and camera poses, it is essential to include SLAM/incremental SfM methods, such as DroidSLAM, ORB-SLAM, DPVO, COMO, anchor-free SfM, VGG-SfM, and COLMAP (in video mode), for fair comparison. The lack of these methods in the discussion, paired with the claim that GLOMAP is a video-based SfM approach, is wired.

**Questions:**

Posted in the weakness.

---

### Official Review · Reviewer_BNhq · 2024-11-02

**Soundness:** 2
**Presentation:** 3
**Contribution:** 2
**Rating:** 5
**Confidence:** 3

**Summary:**

This paper presents a joint approach for video generation and camera estimation. It utilizes intermediate features generated by OpenSora, feeding them into DUSt3R’s camera calibration decoder. By applying both video generation loss and camera reconstruction loss, the model achieves joint optimization, effectively balancing both tasks. Through the integration of 3D camera reconstruction tasks, experimental results demonstrate improved FVD scores, enhancing the overall quality of video generation.
The contributions are summarized as follows: 1. It introduces the first model capable of simultaneously generating videos and estimating 3D camera poses; 2. It provides extensive experimental results on the RealEstate10K-test and DL3DV10K datasets, showcasing the model's effectiveness.

**Strengths:**

- The paper is well-written and easy to follow, providing a clear and coherent narrative throughout.

- The proposed method is straightforward and logically sound, making it accessible for readers to understand its underlying principles.

- The experiments on camera estimation and video generation are thoroughly conducted, effectively demonstrating the method's efficacy.

**Weaknesses:**

- In my view, while the proposed method demonstrates a commendable effort in integrating state-of-the-art video generation and camera estimation models, it may lack the level of novelty one might expect from a significant advancement in the field. The results presented in Tables 1 and 2 do not appear to showcase particularly impressive performance improvements or efficiency gains. For instance, it is worth noting that the DUSt3R* model has already achieved superior estimation results compared to the proposed approach, which raises questions about the distinct contributions of this work.

- Minor typos: Some references are duplicated (e.g., L787-792).

**Questions:**

- Why is the output of the 26th block used as feature maps for the DUSt3R decoders? Would selecting feature maps from a different layer affect performance?

---

### Official Review · Reviewer_uk3j · 2024-11-03

**Soundness:** 2
**Presentation:** 3
**Contribution:** 2
**Rating:** 5
**Confidence:** 5

**Summary:**

This paper proposes to combine video generation with video camera pose estimation. This is achieved by approximating the video DiT backbone in video generation model with the image-based ViT encoder in camera pose estimator DUSt3R, so that the features used to decode a video can simultaneously be used to estimate the relative poses between frames.

**Strengths:**

- paper is easy to follow
- introducing the success of DUSt3R to videos is very useful in many downstream applications and tasks
- I like the observation that ViT and DiT share many architectural designs in common and the follow-up insight of replacing the image-based ViT encoder in DUSt3R with the video DiT backbone in OpenSora

**Weaknesses:**

My major concern lies in the validation of the synergy of two tasks, which is not sufficiently convincing right now. It is important as without this synergy this paper is merely conducting a combination of two existing models.
- At L77-79, authors state *we test if with a limited amount of fine-tuning, one can produce video generator features that also can be reused for camera tracking, without sacrificing video generation quality.* In the experiments authors use all 65K video clips in RealEstate10K, which cannot be considered as limited amount of data or finetuning from my point of view. Considering conducting experiments with varying amounts of fine-tuning data (e.g., 10%, 25%, 50% of RealEstate10K) to demonstrate how performance changes with data size.
- In terms of baselines, authors compare to DUSt3R*, which is trained from scratch as authors stated. From my understanding of the method, JOG3R used the pre-trained decoder of DUSt3R with additional funetuning on RealEstate10K. Therefore, comparing to DUSt3R* makes no sense. JOG3R should be compared to DUSt3R finetuned on RealEstate10k rather than training from scratch.
- Table 2 indicates DUSt3R* performs the best across almost all metrics. This indicates the generalization of JOG3R, in order words, the synergy between two tasks, is not very well. Generalization is important for camera pose esmation.
- For verifying the synergy from camera pose estimation, JOG3R should be compared to pretrained DUSt3R finetuned on RealEstate10k.
- For verifying the synergy from video generation, JOG3R should be compared to pretrained OpenSora finetuned on RealEstate10k, which means in Table 3, JOG3R should be compared to OpenSora finetuned on RealEstate10K rather than the original one.
- a minor point: authors have only evaluated on OpenSora while there are other open-sourced video generation models.

**Questions:**

See weaknesses.

---

### Official Review · Reviewer_fv2F · 2024-11-04

**Soundness:** 3
**Presentation:** 2
**Contribution:** 2
**Rating:** 5
**Confidence:** 4

**Summary:**

The paper propose a framework to enable joint video generation and 3D camera reconstruction. Sepcifically, they utilizes intermediate features of a pre-trained video generation model for predicting relative 3D point maps and hence enabling camera registration. The video generator can generate video along with its camera estimation.

**Strengths:**

The paper’s main theme is presented very clearly and concisely. This straightforward approach aids in understanding the core contributions and objectives of the work.

**Weaknesses:**

1. Clarification on the Benefits of Joint Training
The paper proposes using a diffusion-based video generator to simultaneously produce video frames and camera estimations. However, this approach appears to rely primarily on low-noise conditions (t < 10%), where the input closely resembles a clean video. This raises an important question: how does this approach differ from generating the video first and then applying an existing camera estimation predictor afterward? The advantage of combining these processes in this manner is not immediately clear. Could the authors clarify the benefits of this integration? If the diffusion video generator used alongside the camera estimation predictor merely serves to improve camera estimation performance, it risks being perceived as a simple replacement for the feature backbone rather than a substantial contribution.
2. Lack of Significant Improvement
As shown in Table 1 and Table 2, the proposed method does not demonstrate a significant improvement over deep learning-based DUSt3R or global SfM-based GLOMAP. This lack of substantial performance gain further emphasizes the concern raised in the first question about the added value of integrating video generation and camera estimation.

**Questions:**

1. Clarification on the Benefits of Joint Training
The paper proposes using a diffusion-based video generator to simultaneously produce video frames and camera estimations. However, this approach appears to rely primarily on low-noise conditions (t < 10%), where the input closely resembles a clean video. This raises an important question: how does this approach differ from generating the video first and then applying an existing camera estimation predictor afterward? The advantage of combining these processes in this manner is not immediately clear. Could the authors clarify the benefits of this integration? If the diffusion video generator used alongside the camera estimation predictor merely serves to improve camera estimation performance, it risks being perceived as a simple replacement for the feature backbone rather than a substantial contribution.
2. Lack of Significant Improvement
As shown in Table 1 and Table 2, the proposed method does not demonstrate a significant improvement over deep learning-based DUSt3R or global SfM-based GLOMAP. This lack of substantial performance gain further emphasizes the concern raised in the first question about the added value of integrating video generation and camera estimation.

3. Suggestion on Introduction Section:
The introduction is overly simplistic and lacks sufficient development of the motivation behind the work. Expanding this section to include a deeper exploration of the problem context and the specific challenges or gaps this paper aims to address would enhance the reader’s understanding. A more detailed discussion of the motivation, supported by insights into why this approach is needed and what unique contributions it brings, would strengthen the foundation of the work.

---

### Note · Authors · 2024-11-15

I have read and agree with the venue's withdrawal policy on behalf of myself and my co-authors.